# Age and Sex Influence the Use of Modular Femoral Components in Total Hip Arthroplasty Performed for Primary Osteoarthritis

**DOI:** 10.3390/jcm12030984

**Published:** 2023-01-27

**Authors:** Francesco Castagnini, Barbara Bordini, Monica Cosentino, Michele Di Liddo, Giuseppe Tella, Claudio Masetti, Francesco Traina

**Affiliations:** 1Ortopedia-Traumatologia e Chirurgia Protesica e dei Reimpianti d’Anca e di Ginocchio, IRCCS Istituto Ortopedico Rizzoli, Via Pupilli 1, 40136 Bologna, Italy; 2Laboratorio di Tecnologia Medica, IRCCS Istituto Ortopedico Rizzoli, Via di Barbiano 1/10, 40136 Bologna, Italy; 3Dipartimento di Scienze Biomediche e Neuromotorie (DIBINEM), University of Bologna, 40126 Bologna, Italy

**Keywords:** dual-taper, exchangeable neck, gender, sex-specificity, total hip arthroplasty, young

## Abstract

The impact of age and sex on femoral component choices in modular total hip arthroplasty (THA) is still unknown. A regional arthroplasty registry was interrogated about a modular stem in THA performed for primary osteoarthritis, with the aims to assess the influence of age and sex on stems sizes and neck choices. A total of 6830 THAs were included: all THAs had a modular stem (with 15 necks and 27 combinations per side). Patients were stratified by age in decades and sex. Necks were grouped according to the type of correction. The percentage of larger stem sizes increased in males and in elder patients (*p* < 0.001). Standard necks were overrepresented in males aged 40–59 and underrepresented in males aged 70 or older (*p* < 0.001). Half of the necks provided other corrections than standard or offset, especially in males aged 40–49 and females aged 70 or older (*p* < 0.001). Offset necks were predominant in elder patients (*p* < 0.001). Version-correcting necks were prevalent in younger males and older females (*p* < 0.001). Varus necks were implanted in one-third of the cases. The four commonest necks showed age and sex specific patterns. In the registry, age and sex impacted stem size and neck choices in THA performed for primary osteoarthritis.

## 1. Introduction

Age and sex are two well-known modifiers of the femoral anatomy [1,2,3,4]. Neck–shaft angle and femoral neck version tend to reduce with age, while femoral offset increases [4]. In comparison to females, males tend to have longer necks, larger canal flares, higher offsets, higher neck–shaft angles and lower femoral anteversions [1,4]. These age and sex dependent differences may have an impact on the optimal reconstruction of the hip biomechanics when total hip arthroplasty (THA) is performed, even for primary osteoarthritis [1,4]. As a matter of fact, after THA, an increased femoral lengthening in females and an incomplete acetabular offset reconstruction in men resulted in inferior clinical outcomes in the short term [5,6]. However, whether age and sex dependent differences may practically affect the femoral component choice in THA is substantially unknown.

Conventional single taper stems may not display an adequate sensitivity to detect these differences due to limited proximal femoral choices. It is reported that, in primary hip osteoarthritis, around 30–40% of the femoral anatomies are not matched by conventional single taper stems [1]. In other words, single taper tapered stems with metaphyseal fit designs require a large portfolio of at least 15 sizes, three metaphyseal configurations and two different neck–shaft angles to match the coronal femoral anatomy of only 85% of the femurs [7]. On the contrary, dual taper femoral components with multiple options may closely reproduce the native femoral anatomy: in a study by Duwelius et al., the modular stem cohort (ML Taper Kinectiv, Zimmer, Warsaw, USA) restored femoral offset and leg length in postoperative radiographs more precisely than single taper conventional stems [8]. Thus, dual taper femoral components with additional modularity and several tridimensional reconstructive options may reveal the age and sex dependent changes that influence the femoral component choices in THA.

Large arthroplasty registries may be particularly useful for this purpose, thanks to systematic data collection on sizeable, diversified cohorts. However, to date, large database studies on modular stems have been conducted with the aims of detecting the survival rates and the reasons for revision of dual taper implants with respect to single taper ones in THAs performed for primary osteoarthritis or congenital pathologies [9,10]. Graves et al. investigated the survival rates and reasons for revisions of 9289 dual taper implants in THAs performed for osteoarthritis and compared the modular cohort data to single taper stems: the routine use of dual taper stems in THAs performed for osteoarthritis was discouraged due to the high revision rates [9]. Castagnini et al. proposed a similar comparison in THAs performed for developmental dysplasia of the hip (2787 dual-taper and 3642 single-taper stems): modular implants achieved non-inferior survival rates in the long term [10]. However, these studies focused on the survivals of different modular stems, but no investigation was performed on the effective impact of age and sex on the choice of modular femoral components in THA.

Thus, an arthroplasty registry was interrogated about a single cementless dual-taper stem (the most implanted one in the registry), with a very large portfolio of modular options. We analyzed how the use of a modular stem with several exchangeable neck options in THA performed for primary osteoarthritis was impacted by age and sex, aiming to assess the influence of age and sex based on (1) stem size; (2) neck choice; (3) stem size and neck choice.

## 2. Materials and Methods

RIPO (Registro dell’Implantologia Protesica Ortopedica) is the Emilia–Romagna regional arthroplasty registry, active since January 2000 [10,11]. The registry actively monitors the hip, knee and shoulder arthroplasty procedures performed in the region, providing surveillance over 4.5 million inhabitants [10,11]. RIPO collects data on primary and revision surgeries using a standard form. This form contains data about the clinical conditions of the patients, the devices (batch and code) and the surgical technique (approach and fixation). Data are crosschecked with other databases and discharge forms, achieving a final capture rate of 98% [10,11].

The structure and the capture rate of RIPO is close to the most important national arthroplasty registries [10,11]. Moreover, RIPO is a member of the International Society of the Arthroplasty Registries (ISAR) [12].

The Apta (Adler Ortho, Milan, Italy) stem was selected: this is the most implanted cementless modular stem in the RIPO registry, with several exchangeable neck options. Introduced to the market in 2004, this stem is an anatomic hydroxyapatite coated modular Ti6Al4V device, available in 8 sizes [9,13]. The modular Ti6Al4V neck system (Modula, Adler Ortho, Milan, Italy) allows to independently adjust and fine-tune length, offset and version, by choosing from three options per each parameter: length (varus, neutral, valgus, with 7.5 mm difference), offset (short, medium, long, with 7.5 mm difference) and version (9° anteversion, neutral, 9° retroversion) [9,13]. Every option is identified using colors and letters, as depicted in the figure (Figure 1).

The modular system provides 15 necks and 27 combinations per side, plus another three head–neck options (Figure 1). The offset range is from 28 mm to 54 mm [13,14]. The neck shaft angle is from 123° to 147° [13,14].

RIPO registry was interrogated about cementless THAs performed for primary osteoarthritis with Apta stem (2004–2019). The patients were stratified by sex and age by decades (<40, 40–49, 50–59, 60–69, 70–79, ≥80 years): height, weight and BMI were collected (Table 1).

Percentages of different stem sizes were stratified for sex and age by decade. Necks were grouped according to length, offset and version: the same stratification per age and sex was proposed. Independence between necks, age and sex was calculated. Similar assessments were performed for stem and neck choices together.

Statistical analyses were performed using SPSS software (version 14.0.1, Chicago, IL, USA) JMP, version 12.0.1 (SAS Institute Inc, Cary, NC, USA, 1989–2007) and R software (version, 4.1.2, Wien, Austria) [15]. Demographics and implant-related features were reported as raw data, ranges and percentages. Stratifications by age and sex were proposed. Independence between groups was tested using a Cochran–Mantel–Haenszel test and Pearson’s residuals were plotted using mosaic plots [15,16]. Threshold for significance: *p* = 0.05.

## 3. Results

6830 THAs were suitable for inclusion. Of the total, 59.7% of the implants were implanted in female patients, preferentially in the seventh decade. A total of 1452 THAs (21.3%) were performed in obese patients: 848 implants (20.8%) in females, 604 (22.0%) in males (Table 1).

### 3.1. Stem Size

Percentage of stems sized 6 or larger progressively increased over the decades and were predominant in males (Figure 2).

The dependence between stem sizes, age and sex is depicted in the figure (Figure 3).

### 3.2. Neck Choice

The five most implanted necks were: 0Y red-grey (standard length, no offset, no version, 135°: 1929, 28.2%), 0A green-grey (varus, no offset, no version, 128°: 1360, 19.9%), 9AA green-grey-yellow (varus, no offset, +9° version, 128°: 626, 9.2%), 0C red-black (standard length, +7.5 mm offset, no version, 131°: 520, 7.6%), 9A green-grey-brown (varus, no offset, +9° version, 128°: 480, 7.0%).

The distribution of the necks without specifying the version correction is provided in the table (Table 2).

When the necks were stratified in standard (135° with no offset), offset (131° with 7.5 mm increased lever arm) and other forms of correction, many implants (sometimes even a half) were in the last category (Table 3).

Standard necks were overrepresented in males aged 40–59 and underrepresented in males aged 70 or older (*p* < 0.001, Cochran–Mantel–Haenszel test). Offset necks were underrepresented in females aged 40–69 and overrepresented in females aged more than 80 years and in males aged 70–79 (*p* < 0.001, Cochran–Mantel–Haenszel test). Other forms of correction were more frequent in males aged 40–49 and females aged 70 or older (*p* < 0.001, Cochran–Mantel–Haenszel test).

When necks were stratified according to the version (neutral and ante/retroversion), around one third of the implants provided some correction on the axial plane (Table 4).

In the fourth decade, neutral necks were preferred in males and scarcely adopted in females (*p* < 0.001, Cochran–Mantel–Haenszel test). Neutral necks were over-represented in males on their fifties and females older than 80 years (*p* < 0.001, Cochran–Mantel–Haenszel test). In females between 50–59 years and >80 years, an under-prevalence of necks with version correction was noticed (*p* < 0.001, Cochran–Mantel–Haenszel test).

When the necks were stratified according to the length, varus modularity represented at least one third of the cases, with valgus necks scarcely represented (Table 5).

In males, standard necks were more frequent between 40–49 years and valgus necks were more frequent between 50–59 years (*p* < 0.001, Cochran–Mantel–Haenszel test).

### 3.3. Stem Size and Neck Choice

The distribution of the four commonest neck choices (red-grey, green-grey, green-grey-yellow, red-black) stratified by age, sex and stem size is described in the figure (Figure 4).

The dependence between the four commonest neck choices, the stem size, the age by decades and the sex is depicted in the figures (Figure 5, Figure 6, Figure 7 and Figure 8).

## 4. Discussion

Stem sizes and neck choices were influenced by age and sex. Larger stem sizes were implanted in males and in elder patients. Half of the necks provided corrections others than standard or offset. Offset necks were predominant in elder patients. Version-correcting necks were prevalent in younger males and older females. Varus necks were implanted in one-third of the cases. The four commonest necks showed age- and sex-specific distribution patterns, regardless of the stem sizes.

Age- and sex-driven femoral component choice is still a matter of investigation. While many calculators based on demographic features were developed for component choice in total knee arthroplasty, age and sex (and demographics in general) impacts on femoral choices in THA were far less ascertained [17]. Only a recent paper on 1653 non-modular THAs demonstrated that demographics (age, sex, height, weight and ethnicity) could reliably predict acetabular and femoral component sizes [17]. The authors also developed a calculator with the aim to assist/improve the radiographic planning and streamline operative procedures [17]. However, in addition to some practical consequences on templating and inventory, the wide variability provided by modular femoral components may allow some speculations about the femoral anatomy (and its variations) and the THA component choice.

In this way, the report indirectly confirmed age and sex as two notable modifiers of the femoral anatomy, truly impacting the femoral component choice in daily practice [1,2,3,4,18,19,20]. The wider femoral shafts of males and aged patients in general, noticed by Traina et al. and Sparks et al. in radiological analyses, were also indirectly confirmed by this report: larger stem sizes were preferentially implanted in males and in elder patients (Figure 3) [1,3]. The use of offset necks in aged males confirmed the radiological evidence provided by Carmona et al., noticing longer lever arms in elder patients [4]. On the contrary, the anatomic findings of lower femoral anteversion in aged males did not pair with the neck choice [20]. This finding may be also due to contemporary acetabulum remodeling, which, considering the combined anteversion of the two components, may reduce the impact of femoral anteversion change [20]. Moreover, the wider femoral shaft may allow for some more version adjustment of the stem, larger in younger patients. However, version correcting necks accounted for around one third of the modular implants. A consideration should be made about ante-retroverted necks. The implant involved is an anatomic stem: these devices showed less degree of axial correction than tapered stems, making neck version correction more frequent and desirable [21,22]. It is likely that in tapered stems many necks providing ante-retroversion would be of minor interest: however, this conclusion is merely speculative.

The anatomic finding of varus neck–shaft angle prevalence in aged patients did not significantly impact the neck choice [4,19]. Varus necks were frequently adopted in all decades and sex, showing the importance of varus angulated stems. Specifically, the rate of varus necks reached around one third of the cases in many subgroups and was the second most implanted neck after the standard one (Table 2, Figure 4, Figure 5, Figure 6, Figure 7 and Figure 8). This interest finding may complete the biomechanical evaluations performed by Boese et al. and Carmona et al. [4,23]. Boese et al., who evaluated the reconstruction provided by the three options of Corail stem (DePuy, Warsaw, IN, USA), noticed that three femoral anatomies could not be appropriately addressed by the conventional single-taper stems: low femoral offset and low femoral neck height, varus configuration and isolated high femoral neck height [23]. Even Carmona et al. highlighted a lateralized varus prevalence in males and older patients [4]. Thus, all the studies tended to support the importance of varus alternatives in order to match as many proximal femoral anatomies as possible.

This study has some very strong points. To our knowledge, this is the first report about the use of modular necks stratified for age and sex in a sizeable number of patients, all the implants involving the same stem design (6830 implants, which favorably compared to some of the largest registry papers about modularity) [9,10,11]. This study may give a practical, daily perspective, highlighting which implant is more suitable to restore soft tissue tensioning and implant stability in different age and sex subgroups: some of the limits of anatomic studies are overcome. On the contrary, the study shows some limitations. No anatomic analysis was performed to assess and compare the possible mismatch between anatomy and reconstruction: such an assessment should be performed with specifically designed devices in intraoperative setting [24]. The risk of neck failure may have prevented the surgeons from choosing some hazardous combinations in younger and active patients. In particular, the rate of longer lever arms in male patients below 70 years may have been reduced; similar behavior may have ensued in overweight or obese patients [25]. Some surgeons may have chosen other stem designs with single-taper options in order to avoid the risks of modular neck failures in some specific cases. Moreover, the regional registry could not account for the diversity in femoral anatomies due to climate, lifestyle and ethnicity [15].

## 5. Conclusions

In summary, age and sex impacted the stem and neck choices in THAs performed for primary osteoarthritis in the registry. Larger stems sizes were more frequent in elder and male patients, offset necks were common in elder patients. Mainly, the large adoption of varus and version correcting necks demonstrated that larger portfolios of femoral components in single and dual taper stems may be welcomed even in more conventional anatomies.

## Figures and Tables

**Figure 1 jcm-12-00984-f001:**
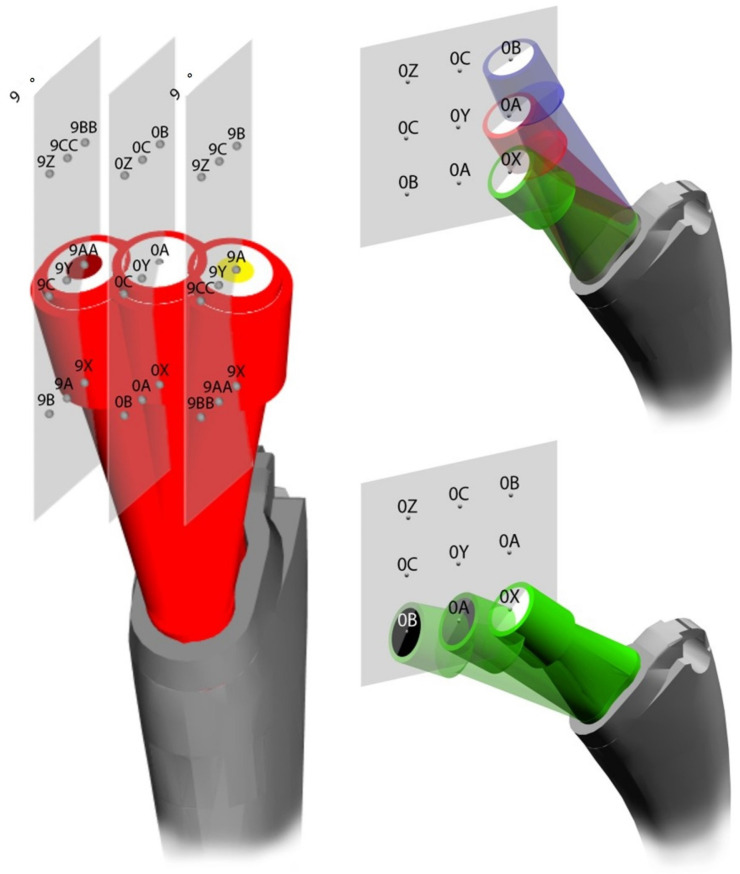
Modula system (Adler Ortho, Milan, Italy) provides different color combinations: green, red and blue for length correction; white, grey and black for offset correction; and yellow and brown for 9° version correction.

**Figure 2 jcm-12-00984-f002:**
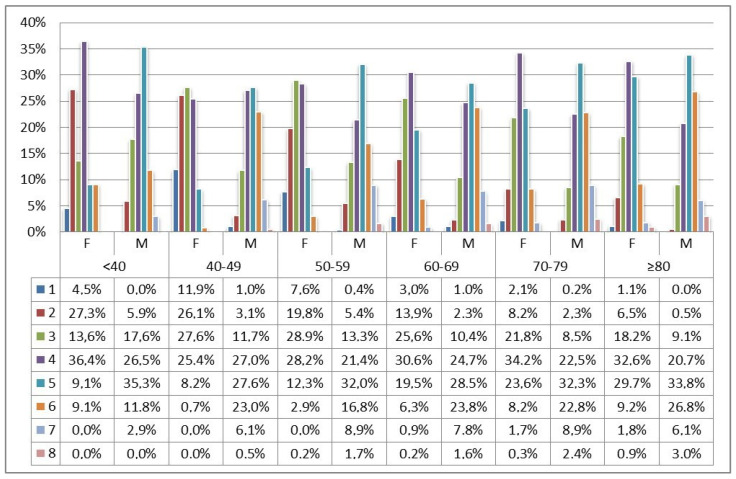
Stem size distribution (1–8) was stratified by age and sex.

**Figure 3 jcm-12-00984-f003:**
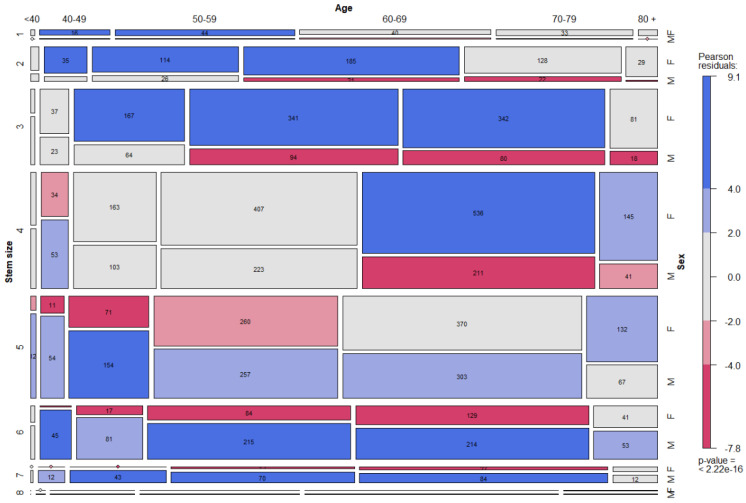
The dependence between stem size, decade and sex is depicted in mosaic plot categories, where categories in red are underrepresented (e.g., females aged 70–79 with stem size 6) and categories in blue are overrepresented (e.g., males aged 70–79 with stem size 6). The width of the columns and the vertical length of the bars are proportional to the number of observations.

**Figure 4 jcm-12-00984-f004:**
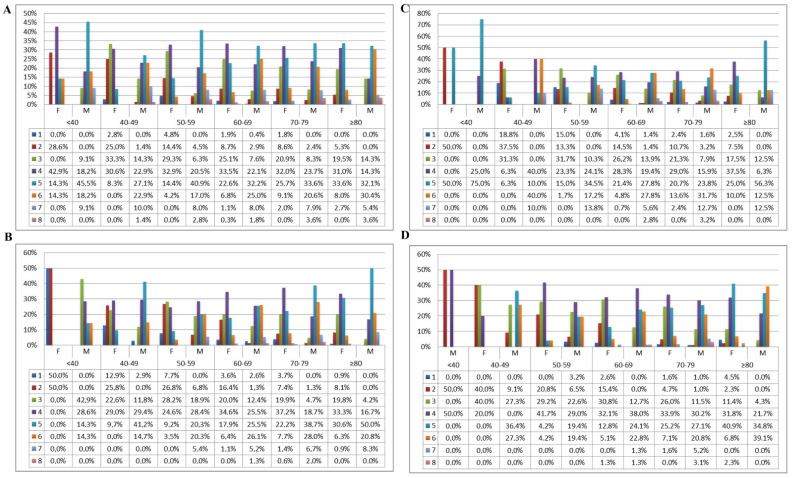
The distribution of the four commonest neck choices was stratified by age, sex and stem size (1–8). (**A**): red-grey; (**B**): green-grey; (**C**): green-grey-yellow; (**D**): red-black.

**Figure 5 jcm-12-00984-f005:**
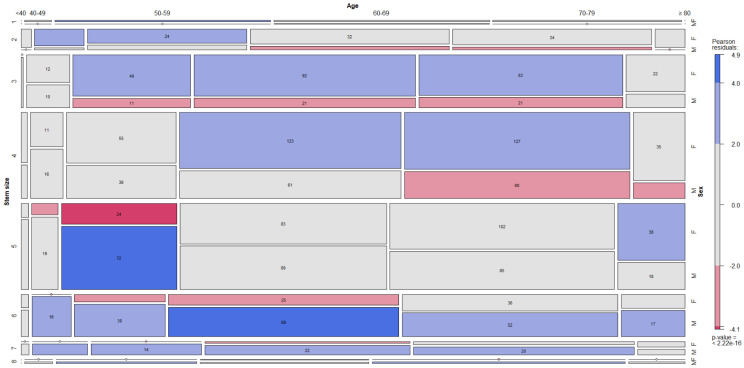
The dependence between red-grey neck choice, stem size, decade and sex is depicted in mosaic plot categories.

**Figure 6 jcm-12-00984-f006:**
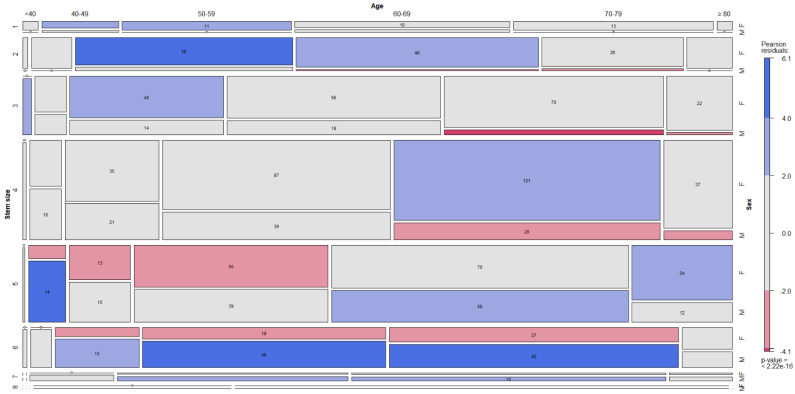
The dependence between green-grey neck choice, stem size, decade and sex is depicted in mosaic plot categories.

**Figure 7 jcm-12-00984-f007:**
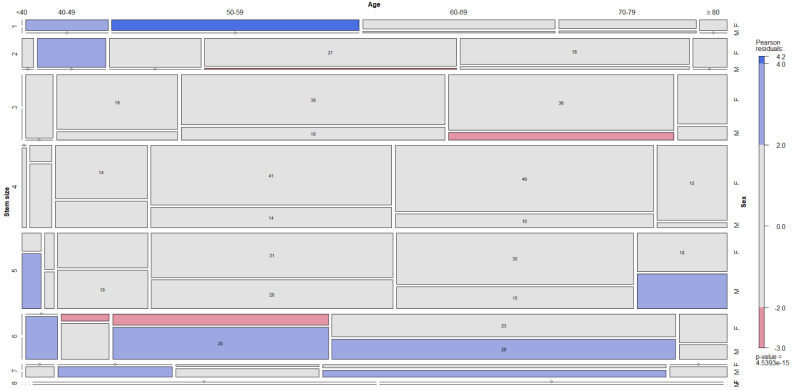
The dependence between green-grey-yellow neck choice, stem size, decade and sex is depicted in mosaic plot categories.

**Figure 8 jcm-12-00984-f008:**
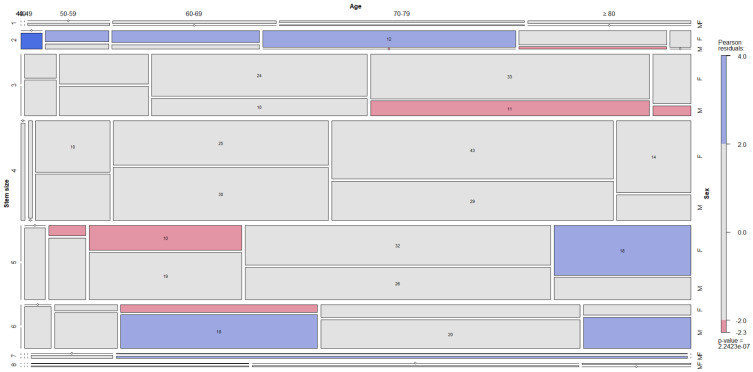
The dependence between red-black neck choice, stem size, decade and sex is depicted in mosaic plot categories.

**Table 1 jcm-12-00984-t001:** Demographics of the modular neck cohorts stratified by age and sex demonstrated a female prevalence.

	Number of THA	Percentage of THA	Mean Weight (kg)	Range Weight (kg)	Mean Height (cm)	Range Height (cm)	Normal BMI (Percentage Females-Men)	Overweight BMI (Percentage Females-Men)	Obese BMI (Percentage Females-Men)
**Female**	4079	59.7	69.5	38–170	161.0	130–184	70.1	53.2	58.4
<40	22	0.5	69.0	49–112	165.6	156–184	0.9	0.2	0.4
40–49	134	3.3	67.9	45–100	163.3	148–180	4.9	2.4	2.6
50–59	577	14.1	71.3	45–125	162.6	140–180	14.3	12.6	16.6
60–69	1332	32.7	70.4	43–170	161.1	137–180	29.9	33.4	35.3
70–79	1569	38.5	69.2	38–120	160.3	130–178	37.0	40.0	39.2
≥80	445	10.9	65.9	41–118	160.0	135–175	12.9	11.4	6.0
**Male**	2751	40.3	82.1	44–142	172.2	145–200	29.9	46.8	41.6
<40	34	1.2	84.1	64–106	176.7	163–195	1.4	1.4	1.0
40–49	196	7.1	84.8	57–128	176.1	155–194	10.0	6.2	7.0
50–59	481	17.5	83.6	52–130	173.6	155–200	15.8	17.9	17.7
60–69	903	32.8	83.2	45–142	172.3	150–199	29.2	32.7	36.4
70–79	939	34.1	80.2	44–130	170.7	145–193	36.1	33.9	32.6
≥80	198	7.2	80.1	59–115	171.1	150–190	7.6	7.9	5.3
**Total**	6830	100.0	74.6	38–170	165.5	130–200	100.0	100.0	100.0

**Table 2 jcm-12-00984-t002:** When necks groups were based on offset and length (regardless the version), percentage distribution by age and sex showed a prevalence of standard and varus devices.

Neck Type	<40		40–49		50–59		60–69		70–79		≥80		
	F	M	F	M	F	M	F	M	F	M	F	M	Total
Blue White	0	0	0	0	0	0	1	0	1	0	0	0	0
Blue Grey	0	3	1	6	3	7	3	4	2	4	2	2	3
Blue Black	0	0	1	3	1	2	1	3	1	2	0	2	1
Red White	0	0	2	1	2	1	3	1	2	1	2	2	2
Red Grey	64	38	43	45	39	46	38	41	34	37	35	37	38
Red Black	0	9	5	9	6	8	7	11	10	14	12	14	11
Green White	0	0	3	0	1	0	1	0	1	1	2	1	1
Green Grey	27	38	41	31	40	28	40	31	41	29	41	30	36
Green Black	9	12	4	6	7	8	7	8	8	11	7	12	8
Total	100	100	100	100	100	100	100	100	100	100	100	100	100

**Table 3 jcm-12-00984-t003:** When necks groups were based on standard and offset configurations (regardless the version), percentage distribution by age and sex showed a prevalence of other corrective modularity.

	<40		40–49		50–59		60–69		70–79		≥80		
	F	M	F	M	F	M	F	M	F	M	F	M	Total
Standard	64	38	43	45	39	46	38	41	34	37	35	37	38
Offset	0	9	5	9	6	8	7	11	10	14	12	14	11
Others	36	53	62	46	55	46	55	48	56	49	53	48	51
Total	100	100	100	100	100	100	100	100	100	100	100	100	100

**Table 4 jcm-12-00984-t004:** When neck groups were based on version, percentage distribution by age and sex showed that one third of ante-retroverted necks were adopted.

	<40		40–49		50–59		60–69		70–79		≥80		
	F	M	F	M	F	M	F	M	F	M	F	M	Total
Neutral	50	71	59	71	59	73	64	67	65	67	69	63	67
Version	50	29	41	29	31	27	36	33	35	33	31	37	33
Total	100	100	100	100	100	100	100	100	100	100	100	100	100

**Table 5 jcm-12-00984-t005:** When necks groups were based on length (blue: valgus; red: standard; green: varus), percentage distribution by age and sex showed a large amount of varus necks.

	<40		40–49		50–59		60–69		70–79		≥80		
	F	M	F	M	F	M	F	M	F	M	F	M	Total
Blue	0	3	1	8	5	9	4	7	4	6	2	4	5
Red	64	47	50	55	47	56	48	53	47	52	49	53	50
Green	36	50	49	37	48	36	48	40	50	41	49	43	45
Total	100	100	100	100	100	100	100	100	100	100	100	100	100

## Data Availability

All the data are publicly available at RIPO (http://ripo.cineca.it/authzssl/index.htm accessed on 23 November 2022).

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
