# Peer review of "Age and Sex Influence the Use of Modular Femoral Components in Total Hip Arthroplasty Performed for Primary Osteoarthritis"

_jcm, 2023, doi:10.3390/jcm12030984_

Round 1
Reviewer 1 Report
1. The abstract section should be enhanced to include quantitative data.
2. Please give a “take-home” message as the conclusion of your abstract.
3. Rearrange keywords alphabetically.
4. Introduction is only just two paragraphs? It needs major development.
5. Describe the novelty of the article made by the author? From the results of my evaluation, it seems that many similar published works adequately explain what you have raised in the current manuscript. If there is something others really new in this manuscript, please highlight it more clearly in the introduction section.
6. Previous study related needs to explain in the introduction section consisting of their work, their novelty, and their limitations to show the research gaps that intend to be filled in the present study.
7. The last paragraph of the introduction section should be the objective of the present study.
8. Restoring the condition of the inflamed hip joint through surgery with total hip replacement is a surgical intervention that is highly effective today. The authors needs to giving the additional explanation in their introduction. Therefore, the MDPI's suggested reverence should be applied in the explanation as follows: Ammarullah, M. I.; Santoso, G.; Sugiharto, S.; Supriyono, T.; Wibowo, D. B.; Kurdi, O.; Tauviqirrahman, M.; Jamari, J. Minimizing Risk of Failure from Ceramic-on-Ceramic Total Hip Prosthesis by Selecting Ceramic Materials Based on Tresca Stress. Sustainability 2022, 14, 13413. https://doi.org/10.3390/su142013413
9. Rather than relying just on the predominate text as it already exists, the authors could incorporate more illustrations as figures in the materials and methods section that illustrate the workflow of the current study.
10. What is the basis for respondent selection? Is there any protocol, standard, or basis that has been followed? It is unclear since the patient is very heterogeneous with a small number. The resonance involved impacts the present result makes this study flaws. One major reason for rejecting this paper.
11. The paper needs to provide critical information on the error and tolerance of the experimental equipment utilized in this work. Due to the disparate outcomes of subsequent research by other researchers, it would make for an insightful conversation.
12. Results must be compared to similar past research.
13. The authors need to improve the discussion in the present article become more comprehensive. The present form was insufficient.
14. Please include the limitation of the present study, it is missing.
15. Where in conclusion section?
16. Mention further research in the conclusion section.
17. The reference needs to be enriched from the literature published five years back. MDPI reference is strongly recommended.
18. The manuscript needs to be proofread by the authors since it has grammatical and language issues.
19. It is mandatory to provide a graphical abstract after revision is submission system.
Author Response
Reviewer 1
- The abstract section should be enhanced to include quantitative data.
R: Quantitative data are included: percentages and p. More specific data should be detailed in the text because are very complex and too long to be detailed in the abstract.
- Please give a “take-home” message as the conclusion of your abstract.
R: There is. “Age and sex impacted stem size and neck choices in THA performed for primary osteoarthritis.”
- Rearrange keywords alphabetically.
R: Done
Changed: Keywords: Dual-taper; Exchangeable neck; Gender; Sex-specificity; Total Hip Arthroplasty; Young.
- Introduction is only just two paragraphs? It needs major development.
R: More developments were added. As age and gender specificity in the choice of femoral component in total hip arthroplasty is a new issue, with minimal literature coverage, the introduction cannot be so long without losing the focus.
Changed: Age and sex are two well-known modifiers of the femoral anatomy [1-4]. Neck-shaft angle and femoral neck version tend to reduce with age, while femoral offset increases [4]. In comparison to females, males tends to have longer necks, larger canal flares, higher offsets, higher neck-shaft angles and lower femoral anteversions, in comparison to fe-males [1,4]. Neck-shaft angles and femoral neck version tend to reduce with age, while femoral offset increases [4]. These age and sex dependent differences may have an impact on the optimal reconstruction of the hip biomechanics when total hip arthroplasty (THA) is performed, even for primary osteoarthritis [1,4]. Warnock et al. noticed that, even if the THA in general achieved satisfying performances, an increased femoral lengthening in females and an incomplete acetabular offset reconstruction in men resulted in inferior clinical outcomes at short-term [5-6]. However, whether these changes age and sex de-pendent differences may practically affect the femoral component stem choice in THA, it is substantially unknown. While Conventional single taper stems may not display an ad-equate sensitivity to detect these differences due to limited proximal femoral choices. It is reported that, in primary hip osteoarthritis, around 30-40% of the femoral anatomies are not matched by conventional single taper stems [1]. On the contrary, dual taper femoral components with additional modularity and wide tridimensional reconstructive options may reveal the age and sex dependent changes that influence the reconstructive femoral parameters and the femoral component choice in THA. However, to date, there is no investigation about the effective impact of age and sex on the choice of modular femoral components in total hip arthroplasty (THA) performed for primary osteoarthritis.
Thus, using an arthroplasty registry, we analyzed how the use of a modular stem with several exchangeable neck options was impacted by age and sex, aiming to assess the influence age and sex on: 1) stem size; 2) neck choice; 3) stem size and neck choice.
- Describe the novelty of the article made by the author? From the results of my evaluation, it seems that many similar published works adequately explain what you have raised in the current manuscript. If there is something others really new in this manuscript, please highlight it more clearly in the introduction section.
R: the article is novel, the analysis of the distribution of modular component per age and sex is unpublished. Moreover, this is an indirect method to analyze if increasing the portfolio of conventional single taper stem can be useful and marketable. The novelty was explained in a sentence just before objectives and was fully discussed in the discussion section, when appropriate comparisons with the current literature were provided.
Changed: However, to date, there is no investigation about the effective impact of age and sex on the choice of modular femoral components in total hip arthroplasty (THA) performed for primary osteoarthritis.
- Previous study related needs to explain in the introduction section consisting of their work, their novelty, and their limitations to show the research gaps that intend to be filled in the present study.
R: see above. Discussed in the other points.
- The last paragraph of the introduction section should be the objective of the present study.
R: done
Changed: Thus, using an arthroplasty registry, we analyzed how the use of a modular stem with several exchangeable neck options was impacted by age and sex, aiming to assess the influence age and sex on: 1) stem size; 2) neck choice; 3) stem size and neck choice.
- Restoring the condition of the inflamed hip joint through surgery with total hip replacement is a surgical intervention that is highly effective today. The authors needs to giving the additional explanation in their introduction. Therefore, the MDPI's suggested reverence should be applied in the explanation as follows: Ammarullah, M. I.; Santoso, G.; Sugiharto, S.; Supriyono, T.; Wibowo, D. B.; Kurdi, O.; Tauviqirrahman, M.; Jamari, J. Minimizing Risk of Failure from Ceramic-on-Ceramic Total Hip Prosthesis by Selecting Ceramic Materials Based on Tresca Stress. Sustainability 2022, 14, 13413. https://doi.org/10.3390/su142013413
R: The reference was added
- Rather than relying just on the predominate text as it already exists, the authors could incorporate more illustrations as figures in the materials and methods section that illustrate the workflow of the current study.
R: There are 5 figures, and some of them have more subfigures inside. I would not add any more figures. As there is no complex workflow, but much of the study is a statistical analysis of a large well-defined sample, it is not possible to illustrate the workflow of the study
- What is the basis for respondent selection? Is there any protocol, standard, or basis that has been followed? It is unclear since the patient is very heterogeneous with a small number. The resonance involved impacts the present result makes this study flaws. One major reason for rejecting this paper.
R: A sample of 6830 identical implants cannot be defined a small number at all. This is the largest sample of the same modular implants ever published. Just to give you an example, the Australian registry published a study on modular necks including 9289 femoral stems with exchangeable necks, and several stem designs were included, with different neck alloys. There can be drawbacks and problems with the paper, but not small number and heterogeneity (See Graves SE, de Steiger R, Davidson D, Donnelly W, Rainbird S, Lorimer MF, Cashman KS, Vial RJ. The use of femoral stems with exchangeable necks in primary total hip arthroplasty increases the rate of revision. Bone Joint J. 2017 Jun;99-B(6):766-773).
About the selection process: the Apta stem was chosen as it was the most represented in the registry, has a very large portfolio of modular necks, it is ana anatomic stem with tends to follow the femoral canal and thus requires also ante-retroverted necks. So, it is the most informative stem you can use if you want to analyze age and sex dependent variations.
Changed: Apta (Adler Ortho, Milan, Italy) stem was selected: this is the most implanted cementless modular stem in the RIPO registry, with several exchangeable neck options.
- The paper needs to provide critical information on the error and tolerance of the experimental equipment utilized in this work. Due to the disparate outcomes of subsequent research by other researchers, it would make for an insightful conversation.
R: The info about registry were provided. The stem was described and the modular system was described and represented in figure 1. See Materials and Methods
- Results must be compared to similar past research.
R: Comparisons were provided in the discussion
- The authors need to improve the discussion in the present article become more comprehensive. The present form was insufficient.
R: Discussion was implemented. See Discussion
- Please include the limitation of the present study, it is missing.
R: The limitations are stated.
See the text: Line 231 This study has some very strong points. To our knowledge, this is the first report about the use of modular necks stratified for age and sex in a sizeable number of patients, all the implants involving the same stem design. This study may give a practical, daily perspective, highlighting which implant is more suitable to restore soft tissue tensioning and implant stability in different age and sex subgroups: the limits of anatomic studies are definitively overcome. On the contrary, the study shows some limits. No anatomic analysis was performed to assess and compare the possible mismatch between anatomy and reconstruction: such an assessment should be performed with specifically designed devices in intra-operative setting [20]. The risk of neck failure may have prevented the surgeons from choosing some hazardous combinations in younger and active patients. Especially, the rate of longer lever arms in male patients below 70 years may have been reduced. Some surgeons may have chosen other stem designs with single-taper options in order to avoid the risks of modular neck failures in some specific cases. Moreover, the re-gional registry could not account for the diversity in femoral anatomies due to climate, lifestyle, ethnicity [125].
- Where in conclusion section?
R: Added
Changed: 5. Conclusion
In summary, age and sex impacted the stem and neck choices in THAs performed for primary osteoarthritis. Larger stems sizes were more frequent in elder and male patients, offset necks were common in elder patients. Mainly, the large adoption of varus and ver-sion corrective necks demonstrated that larger portfolios of femoral components may be welcomed even in more conventional anatomies.
- Mention further research in the conclusion section.
R: Pertinent researches were mentioned in the discussion. Bibliography was improved
- The reference needs to be enriched from the literature published five years back. MDPI reference is strongly recommended.
R: Done
- The manuscript needs to be proofread by the authors since it has grammatical and language issues.
R: Done
- It is mandatory to provide a graphical abstract after revision is submission system.
R: provided
Reviewer 2 Report
Very relevant and interesting work,
Some figures must be updated as the axis are not clear, others require some small modifications by adding info (labels) inside the tables. All textual comments can be found in the attached Pdf

Author Response
Reviewer 2
Thank you for your time and your efforts to improve the paper.
- Very relevant and interesting work,
Some figures must be updated as the axis are not clear, others require some small modifications by adding info (labels) inside the tables. All textual comments can be found in the attached Pdf
R: Thank you. Here below are the changes/rebuttals.
- These changes. Please clarify or do you refer to these differences? line 46
R: differences. However, the intro was revised
- 4,500,000 4.5 million line 60
R: revised.
Changed: providing surveillance over 4.5 million inhabitants, through 68 Orthopedic facilities
- Considering splitting this sentence into two separate ones. Second part starts with. This form contains data of. Line 61
R: Revised
Changed: This form contains data about the clinical conditions of the patients, the devices (batch and code) and the surgical technique (approach and fixation).
- Ok why do we need to know this? Please explain why you state this (show that this data can be trusted? Or is validated?) Line 65
R: I think it is important, because ISAR membership needs some basic requirements that makes a registry trusted. I would leave the sentence
- Please explain why only this one is selected as thee are more modular systems that are sold well (depuys?)
R: The stem was selected for several reasons: first, as written, this is the most implanted cementless stem in our registry. We are talking about 6800 Apta stems: this is a very sizeable cohort. Second, the stem relies on a complex modular neck system, that covers a wide range of options: thus it may account for more variability. Moreover, this is an anatomic stem. The anatomic stem tends to follow the femoral canal and does not allow for large version adjustment and requires specific ante-retroverted necks (differently from S-ROM by DePuy). Another option that could be used would have been ABG or ML taper or Metha, which are not represented or are scarcely represented in our registry (and are not so abundant worldwide)
Changed: Apta (Adler Ortho, Milan, Italy) stem was selected: this is the most implanted cementless modular stem in the RIPO registry, with several exchangeable neck options.
- Is this how you ref to a figure? Please check Line 76
R: removed, it was a biblio reference but it is out of place
- Is age category? Not a better term? Line 84
R: A little bit difficult. Rephrased as follows. However, the term decade was reduced and changed with age
Changed: The patients were stratified for sex and age by decades (<40, 40-49, 50-59, 60-69, 70-79, ≥80 years):
- Change labels about BMI in the first table
R: Done
Changed: see table 1
- Not needed..is probably more clear. Otherwise it seems that you applied but the application was waiwed. Please check if this is correct. Line 92
R: It is correct. But it is superfluous and it is specified in the declarations at the end of the manuscript. Removed.
- P=0.05 line 100
R: Done
Changed: Threshold for significance: p=0.05.
- Please check if this is the correct way to refer to a table
R: Changed all over the text, tables and figures
- Colors miss legenda. Please add what do 1-8 mean. Figure 2
R: Stem sizes. Added in the figure legend, and not in the figure, for simplicity and figure readability
Changed: Stem size distribution (1-8) was stratified by decades and sex
- This is very unclear, why are the blocks under the first row shifted to cover different age catagories? you expect that the blocks have the length of all preset age categories (<40, 40-49, 50-59, 60-69, 70-79, ≥80 years). Please consider correction or explain why you deviate from this. This figure must be updated for the reader to understand what age range fit to each block. Figure 3
R: This is a mosaic plot so the width of the columns is proportional to the number of observations in each level of the variable plotted on the horizontal axis and the vertical length of the bars is proportional to the number of observations in the second variable within each level of the first variable. It was summarized in the legend. Not easy to understand if you are not used to, but it is the only statistical way to represent it.
Changed: The dependence between stem size, decade and sex is depicted in mosaic plot categories, where categories in red are under-represented (e.g. females aged 70-79 with stem size 6) and categories in blue are over-represented (e.g. males aged 70-79 with stem size 6). The width of the columns and the vertical length of the bars are proportional to the number of observations.
- Add label above this column (neck type) Table 3
R: Done
Changed: see table 3
- please also add a label "neck size" above the colors 1-8 in each of the 4 sub figures. Figure 4
R: 1-8 is the stem size. As above, I would prefer to avoid it for simplicity and figure readability. Added in the figure legend
Changed: The distribution of the four commonest neck choices was stratified by decades, sex and stem size (1-8).
- These interesting finding may complete...Line 194
R: Done
Changed: This interest finding may be of interest, and may complete the biomechanical evaluations performed by Boese et al. and Carmona et al. [4,15].
- This is a very interesting limitation. Could you also elaborate on the importance of Force and position tracking in the stem so this qualitative data can be used in a database such as REPO to improve the success rate of the procedure and insight in the functioning of the chosen configuration during the ROM tests? [Wei, J. C., Blaauw, B., Van der Pol, D. G., Saldívar, M. C., Lai, C. F., Dankelman, J., & Horeman, T. (2022). Design of an affordable, modular implant device for soft tissue tension assessment and range of motion tracking during total hip arthroplasty. IEEE Journal of Translational Engineering in Health and Medicine.]. Line 208
R: added
Changed: No anatomic analysis was performed to assess and compare the possible mismatch between anatomy and reconstruction, as such an assessment should be performed with specifically designed devices in intra-operative setting [16].
- maybe good to convert into a conclusion if the journal allows this? Line 216
R: Done. Added a section
Round 2
Reviewer 1 Report
Good job to the authors, but I have several issues that needs to be addressed.
1. Line 40-64 in the introduction section just only contain 2 paragraphs? It is serious? Major development is mandator. Please consider my previous comments regarding the novel and previous study to enhance the explanation quality. The present quality sill not inappropriate.
2. For authors response for my comments number 10. Its explanation should be included to rebut my previous comments. The sentence mentioning “9289 femoral stems” from the published literature not discussed in the revised version.
3. The authors need to explain the potential further study in computational simulation approach (in silico) that bring several advantages compared to clinical study (in vivo) such as lower cost and faster results. For supporting the explanation, additional reference needs to be incorporated as follows: Ammarullah, M. I.; Santoso, G.; Sugiharto, S.; Supriyono, T.; Kurdi, O.; Tauviqirrahman, M.; Winarni, T. I.; Jamari, J. Tresca Stress Study of CoCrMo-on-CoCrMo Bearings Based on Body Mass Index Using 2D Computational Model. Jurnal Tribologi 2022, 33, 31–8. https://jurnaltribologi.mytribos.org/v33/JT-33-31-38.pdf
Author Response
1. Line 40-64 in the introduction section just only contain 2 paragraphs? It is serious? Major development is mandator. Please consider my previous comments regarding the novel and previous study to enhance the explanation quality. The present quality sill not inappropriate.
R: the intro has been developed, almost doubled (40-86). It roughly accounts for a page.
Previous studies were described, and the novelty of the paper was more strongly introduced.
Changed: 54-86
Conventional single taper stems may not display an adequate sensitivity to detect these differences due to limited proximal femoral choices. It is reported that, in primary hip osteoarthritis, around 30-40% of the femoral anatomies are not matched by conventional single taper stems [1]. In other words, single taper tapered stems with metaphyseal fit designs require a large portfolio of at least 15 sizes, three metaphyseal configurations and two different neck-shaft angles to match the coronal femoral anatomy of only 85% of the femurs [7]. On the contrary, dual taper femoral components with multiple options may closely reproduce the native femoral anatomy: in a study by Duwelius et al., the modular stem cohort (ML Taper Kinectiv, Zimmer, Warsaw, US) restored femoral offset and leg length in post-operative radiographs more precisely than single taper conventional stems [8]. Thus, dual taper femoral components with additional modularity and wide tridimensional reconstructive options may reveal the age and sex dependent changes that influence the femoral component choices in THA.
Large arthroplasty registries may be particularly useful for this purpose, thanks to systematic data collection on sizeable, diversified cohorts. However, to date, large database studies on modular stems have been conducted with the aims to detect the survival rates and the reasons for revision of dual taper implants with respect to single taper ones, in THAs performed for primary osteoarthritis or congenital pathologies [9,10]. Graves et al. investigated the survival rates and reasons for revisions of 9289 dual taper implants in THAs performed for osteoarthritis and compared the modular cohort data to single taper stems: the routine use of dual taper stems in THAs performed for osteoarthritis was discouraged due to the high revision rates [9]. Castagnini et al. proposed a similar comparison in THAs performed for developmental dysplasia of the hip (2787 dual taper and 3642 single taper stems): modular implants achieved non-inferior survival rates at long-term [10]. However, these studies focused on the survivals of different modular stems, but no investigation was performed about the effective impact of age and sex on the choice of modular femoral components inTHA.
Thus, an arthroplasty registry was interrogated about a single cementless dual taper stem (the most implanted one), with a very large portfolio of modular options. We analyzed how the use of a modular stem with several exchangeable neck options in THA performed for primary osteoarthritis was impacted by age and sex, aiming to assess the influence of age and sex on: 1) stem size; 2) neck choice; 3) stem size and neck choice.
2. For authors response for my comments number 10. Its explanation should be included to rebut my previous comments. The sentence mentioning “9289 femoral stems” from the published literature not discussed in the revised version.
R: This issue has been developed in the intro and in the limitations of the study. The reference was added and other pertinent references were added and discussed.
Changed: 71-77 Graves et al. investigated the survival rates and reasons for revisions of 9289 dual taper implants in THAs performed for osteoarthritis and compared the modular cohort data to single taper stems: the routine use of dual taper stems in THAs performed for osteoarthritis was discouraged due to the high revision rates [9]. Castagnini et al. proposed a similar comparison in THAs performed for developmental dysplasia of the hip (2787 dual taper and 3642 single taper stems): modular implants achieved non-inferior survival rates at long-term [10].
254-257 To our knowledge, this is the first report about the use of modular necks stratified for age and sex in a sizeable number of patients, all the implants involving the same stem design (6830 implants, which favorably compared to some of the largest registry papers about modularity) [9-11].
3 . The authors need to explain the potential further study in computational simulation approach (in silico) that bring several advantages compared to clinical study (in vivo) such as lower cost and faster results. For supporting the explanation, additional reference needs to be incorporated as follows: Ammarullah, M. I.; Santoso, G.; Sugiharto, S.; Supriyono, T.; Kurdi, O.; Tauviqirrahman, M.; Winarni, T. I.; Jamari, J. Tresca Stress Study of CoCrMo-on-CoCrMo Bearings Based on Body Mass Index Using 2D Computational Model. Jurnal Tribologi 2022, 33, 31–8. https://jurnaltribologi.mytribos.org/v33/JT-33-31-38.pdf
R: the reference was added.